# Single-cell profiling identifies pre-existing CD19-negative subclones in a B-ALL patient with CD19-negative relapse after CAR-T therapy

Tracy Rabilloud[1,3], Delphine Potier [1,3], Saran Pankaew[1], Mathis Nozais [1], Marie Loosveld [1,2,4✉] & Dominique Payet-Bornet [1,4✉]

Chimeric antigen receptor T cell (CAR-T) targeting the CD19 antigen represents an innovative therapeutic approach to improve the outcome of relapsed or refractory B-cell acute lymphoblastic leukemia (B-ALL). Yet, despite a high initial remission rate, CAR-T therapy ultimately fails for some patients. Notably, around half of relapsing patients develop CD19 negative (CD19neg) B-ALL allowing leukemic cells to evade CD19-targeted therapy. Herein, we investigate leukemic cells of a relapsing B-ALL patient, at two-time points: before (T1) and after (T2) anti-CD19 CAR-T treatment. We show that at T2, the B-ALL relapse is CD19 negative due to the expression of a non-functional *CD19* transcript retaining intron 2. Then, using single-cell RNA sequencing (scRNAseq) approach, we demonstrate that CD19neg leukemic cells were present before CAR-T cell therapy and thus that the relapse results from the selection of these rare CD19neg B-ALL clones. In conclusion, our study shows that scRNAseq profiling can reveal pre-existing CD19neg subclones, raising the possibility to assess the risk of targeted therapy failure.

---

[1] Aix Marseille Université, CNRS, INSERM, CIML, Marseille, France. [2] APHM, Hôpital La Timone, Laboratoire d'Hématologie, Marseille, France. [3]These authors contributed equally: Tracy Rabilloud, Delphine Potier. [4]These authors jointly supervised this work: Marie Loosveld, Dominique Payet-Bornet. ✉email: loosveld@ciml.univ-mrs.fr; payet@ciml.univ-mrs.fr

B-cell acute lymphoblastic leukemia (B-ALL) is a hemato-logic malignancy arising from uncontrolled proliferation of B lymphoblasts. Although the 5-year overall survival rate reaches currently 80–90% for children, the prognosis of relapsed and refractory B-ALL is dismal, and those B-ALL remain the principal cause of cancer-related childhood mortality[1,2]. Hence, novel therapeutic options for these B-ALL cases are urgently warranted. In this context, one of the most innovative approach relies on chimeric antigen receptors (CARs)-T cell therapy[3–6]. CARs are artificial receptors including an extracellular antibody-derived domain recognizing a specific target fused to intracellular signaling domains derived from T cells[7]. Autologous T cells engineered to express CARs (CAR-T) acquire the ability to recognize and to kill efficiently cells harboring at their surface the target antigen. Regarding the treatment of relapsed/refractory B-ALL, adoptive T cells expressing a CAR directed against the B-cell-associated surface marker, CD19, have been used in several clinical trials[8–12]. Those studies showed that CAR-T therapies achieved high remission rates ranging from 70 to 90% at short-term. However, ~40% of responding B-ALL patients ultimately relapse. Two modes of relapse were observed, CD19-positive (CD19$^{pos}$) and CD19-negative (CD19$^{neg}$). The CD19$^{pos}$ relapse mainly results from the low potency and poor persistence of CAR-T cells[13]. Nevertheless, to overcome these issues, efforts are made to optimize clinical strategies and CAR-T cells engineer-ing[14,15]. In CD19$^{neg}$ relapses, which account for around half of the cases (39% and 68% in respectively Gardner et al.[9] and Maude et al.[8] studies), leukemic cells lose CD19 epitope surface expres-sion and thus escape CAR-mediated recognition, inhibiting B-ALL clearance. Recent studies showed that CD19 negativity results mainly from impaired expression of CD19 mRNA (through de novo frameshift/missense CD19 mutations, alter-natively spliced CD19 mRNA species and hemizygous deletions spanning the CD19 locus)[16–19]. However, no study has clearly demonstrated whether CAR-T cell therapy is directly responsible of CD19 transcription dysregulation or simply allows the emer-gence of minor CD19$^{neg}$ clones escaping CD19-targeted therapy. In the second hypothesis, detection of those CD19$^{neg}$ clones may represent a valuable predictor of CD19-negative relapse permit-ting to assess beforehand the risk for CAR-T treatment failure and to adapt accordingly the therapeutic strategy.

Here, we report the single-cell RNA-sequencing (scRNAseq) profiling of leukemic samples from a B-ALL patient, at two-time points, before and after CAR-T therapy. Our work reveals the presence of preexisting CD19$^{neg}$ B-ALL clones before the CAR-T treatment.

## Results

### Analysis of B-ALL samples prior and after CAR-T cell treat-ment. We used scRNAseq approach to investigate leukemic cells of a B-ALL patient at two-time points; before and after anti-CD19 CAR-T cell therapy. The B-ALL patient underwent a relapse after chemotherapy and CD19$^{pos}$ leukemic cells from bone marrow (BM) were harvested (T1 cells = before CAR-T cell therapy; Supplementary Fig. 1a). Then, the patient was treated with anti-CD19 CAR-T. After an initial complete remission, the patient experienced a frank relapse, as noted by circulating blasts and an infiltration (>90%) of CD10$^{pos}$CD19$^{neg}$ leukemic cells in the BM (Supplementary Fig. 1b) from which cells were harvested (T2 cells = after CAR-T cell therapy). Therefore, we asked whe-ther these CD19$^{neg}$ B-ALL clones were present before CAR-T cell therapy. To address this issue, T1 and T2 samples were sorted according to forward scatter and side scatter, a cell viability marker, CD3 and CD19 surface expression (Fig. 1a). We obtained

four tubes of sorted cells: T1-CD19$^{pos}$, T1-CD19$^{neg}$, T2-CD19$^{pos}$, and T2-CD19$^{neg}$. Each of those samples were marked with a distinct anti-CD45-hashtag oligonucleotide (HTO) antibody[20] (Supplementary Table 1), then mixed and analyzed by scRNAseq using the 5′ 10× Genomics technology[21] (Fig. 1a). After libraries sequencing and data preprocessing, we applied the Seurat graph-based clustering algorithm and identified six main clusters. Then, we used the UMAP nonlinear dimensionality reduction method, to visualize cell transcriptome heterogeneity[22]. According to various gene markers, we assigned cell type to clusters (Fig. 1b, c). Non-B cells, i.e., myeloid and NK cells are located in clusters 2 and 5, respectively. Physiological B cells are in clusters 3 and 4, while B-ALL cells are in clusters 0 and 1. Sample demultiplexing assigned 1189, 524, 764, and 442 cells to T1-CD19$^{pos}$, T1-CD19$^{neg}$, T2-CD19$^{pos}$, and T2-CD19$^{neg}$ samples, respectively, and allowed us to visualize sample of origin for each cell on the UMAP plot. Notably, T1-CD19$^{pos}$ and T2-CD19$^{neg}$ tumoral cells were detected mainly to clusters 0 and 1, respectively (Fig. 1d). CaSpER tool[23] and B-allele frequency (BAF) analysis indicate that cells from these two clusters displayed common copy number variants (CNV), such as chromosome 9q deletion, indicating that CD19$^{pos}$ and CD19$^{neg}$ B-ALL are related (Supplementary Fig. 2). The most differentially expressed transcription factor between clusters 0 and 1 was KLF6 gene, a well-known tumor suppressor gene[24,25] (Supplementary Fig. 3). Strikingly, CD19 mRNA transcripts were detected in T2-CD19$^{neg}$ tumoral cells albeit at lower level than in T1-CD19$^{pos}$ tumoral cells (Supplementary Fig. 4a, b). The analysis of bulk cDNA from T1-CD19$^{pos}$ and T2-CD19$^{neg}$ sorted cells shows that T2-CD19$^{neg}$ cells do not express full-length transcripts (Fig. 1e). We observed that both samples express a previously described[18] nonfunctional CD19 transcript retaining intron 2 (Supplementary Fig. 4c). However, in T2-CD19$^{neg}$ cells only this nonfunctional isoform is expressed, explaining the absence of CD19 protein despite the presence of CD19 mRNA.

### Detection of preexisting CD19$^{neg}$ B-ALL cells. Importantly, in the UMAP plot, we clearly distinguish in tumoral clusters, 20 cells belonging to T1-CD19$^{neg}$ sample (Figs. 1d and 2a, red dots) indicating that those cells are CD19$^{neg}$ B-ALL. Yet, to further establish that those cells were not mis-assigned, we compared the gene expression profiles of B-ALL cells versus physiological B cells, myeloid or NK cells. Resulting heatmaps of the most differentially expressed genes (Fig. 2b and Supplementary Fig. 5), as well as dot plot for marker genes expression (Fig. 2c) show that these 20 T1-CD19$^{neg}$ cells are alike B-ALL cells. Moreover, CaSpER tool succeeded to detect 9q deletion for half of the 20 T1-CD19$^{neg}$ cells (Supplementary Fig. 2a). In the same line, BAF analysis of RPS14 SNP gave a similar result (loss of heterozygosity of RPS14 allele) for T1-CD19$^{neg}$ B-ALL and the other T1/T2 B-ALL cells (Supplementary Fig. 2b). Altogether, these data confirm that T1-CD19$^{neg}$ cells in B-ALL clusters are true leukemic cells. However, as CD19 negativity was defined by FACS during cell sorting, we validated this observation at the molecular level. Thus, the 20 putative CD19$^{neg}$ B-ALL clones were backtracked using their cell identifier 10× barcode (BC), and CD19 mRNA expression was assessed by nested-PCR (Fig. 2d). As shown in Fig. 2e, one specific CD19 mRNA band is clearly detected for the three positive control cells (CD19$^{pos}$ B-ALL from T1-CD19$^{pos}$ samples), conversely only background levels of various PCR products were observed for negative control cells (CD19$^{neg}$ B-ALL from T2-CD19$^{neg}$ samples). Among the 20 CD19$^{neg}$ B-ALL clones spotted at T1, 10 clones were clearly similar to control CD19$^{neg}$ B-ALL cells (i.e., background level

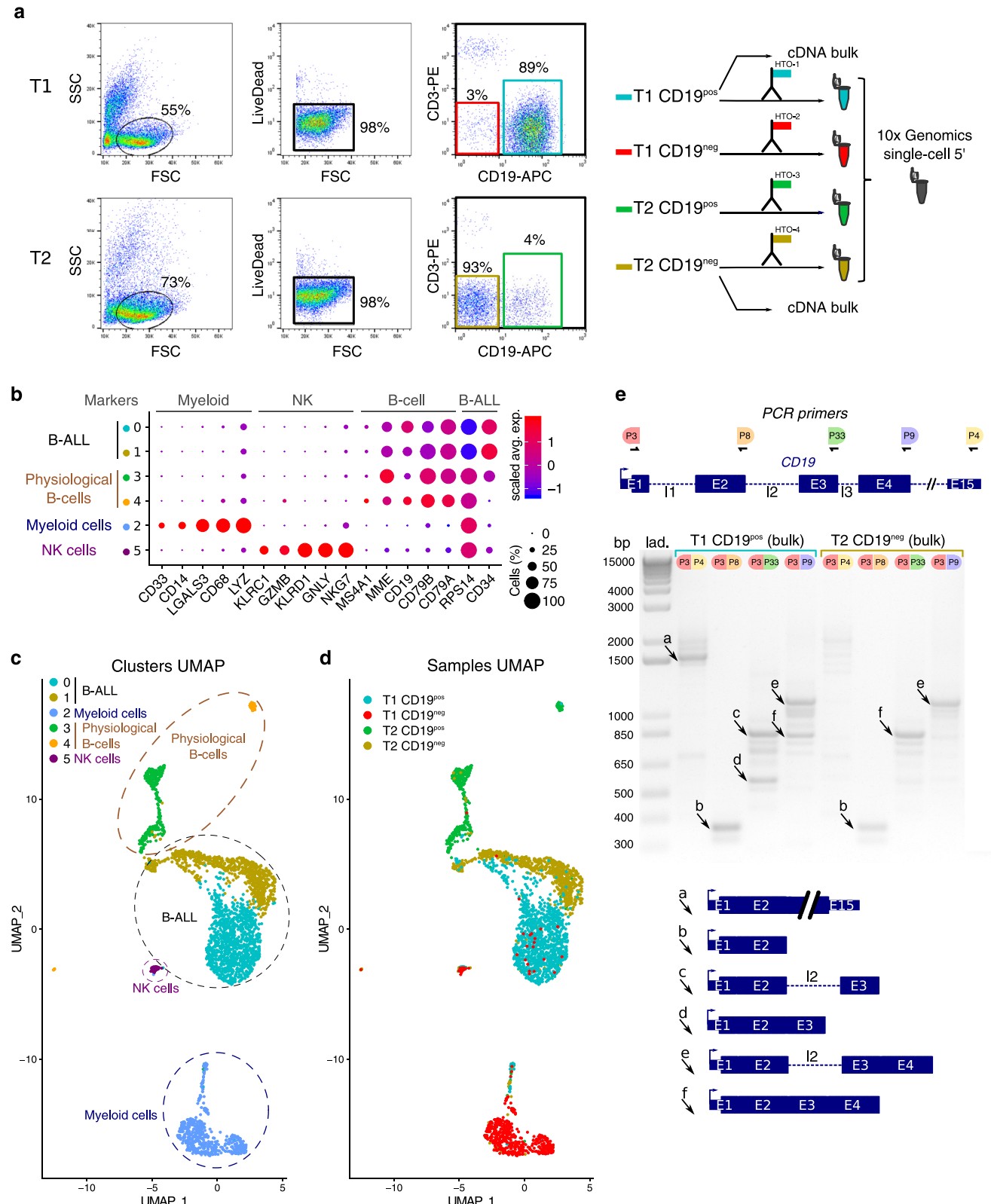

of PCR products and absence of a clear specific CD19 mRNA band; Fig. 2e). The PCR profiles of the remaining ten clones were ambiguous, as a faint CD19 mRNA band was detected (Supplementary Fig. 6). Overall, our results demonstrate that CD19neg B-ALL clones were present before anti-CD19 CAR-T therapy, which, per se, is thus not responsible for CD19 mRNA dysregulation.

## Discussion

In addition to somatic mutations found in exons 2–5 (ref. [16]), mis-splicing of *CD19* mRNA is an alternative way to impede CD19 expression at the cell surface[18,19]. In this context, Sotillo et al.[19] showed that skipping of *CD19* exon 2 is due to down-regulation of SRSF3 splicing factor, which is normally involved in exon 2 retention. Herein, our analysis of CD19 sequencing reads

**Fig. 1 CD19[neg] B-ALL relapse following CAR-T therapy. a** Cell sorting strategy. Cells before (T1) and after (T2) CAR-T treatment were gated according to FSC/SSC profile (left FACS plots). Then, live cells (middle FACS plots) were analyzed according to CD3 and CD19 expression (right FACS plots; see also Supplementary Fig. 9). Gates of the four sorted subpopulations T1-CD19[pos], T1-CD19[neg], T2-CD19[pos], and T2-CD19[neg] are highlighted in cyan, red, green, and gold, respectively. Those sorted subpopulations were labeled with a specific anti-CD45-HTO antibody, then multiplexed and analyzed by scRNAseq using the 10× Genomics single-cell 5′ technology. Some cells from T1-CD19[pos] and T2-CD19[neg] samples were used in bulk to prepare cDNA. **b** Seurat Dotplot showing the expression level of marker genes in each cluster. Dot size represents the percentage of cell expressing the gene of interest, while dot color represents the scaled average expression (Scaled Avg. Exp.) of the gene of interest across the various clusters (a negative value corresponds to an expression below the mean expression). We used CD34 and RPS14 expression as tumoral markers. Indeed FISH analysis revealed a 5q32 deletion, explaining RPS14 lower expression in B-ALL cells. **c** UMAP visualization of the six main clusters and their corresponding cell types of T1 and T2 sorted samples. **d** UMAP visualization of the four demultiplexed samples: T1-CD19[neg], T1-CD19[pos], T2-CD19[neg], and T2-CD19[pos]. **e** Agarose gel of CD19 cDNA amplified products using exons-specific primer sets depicted on the top panel. PCR were performed with bulk cDNA from T1-CD19[pos] cells and T2-CD19[neg] cells. Agarose gel data are representative of two independent experiments. Lane "Lad" is the 1 kb DNA size marker. Schematic representations of PCR products indicated by "a"–"f" arrows are shown below the gel.

did not reveal any somatic mutation within exons 2–5. However, we showed that loss of CD19 is due to the expression of a non-functional *CD19* isoform retaining intron 2. This alteration was previously described as a recurrent event[18], yet further investigations would be required to characterize the mechanism of intron 2 retention.

Here, using scRNAseq approach, we detected CD19[neg] B-ALL cells before CAR-T treatment. A previous study suggested the presence of such preexisting clones[6]. Yet, in this study CD19[neg] cells (~7%) were detected by FACS after a gating strategy based on live cells that were CD45[pos], SSC low, and CD34[pos] (ref. [6]). Thus, this analysis did not formally exclude some non-B-ALL cells, such as myeloblasts. We attempted to identify by FACS CD19[neg] B-ALL clones; however, those potential clones represent only 0.03% of live cells and were not clearly clustered (Supplementary Fig. 7), and thus require molecular biology to convincingly validate their leukemic origin.

In conclusion, our study demonstrates that a Darwinian-like selection of preexisting CD19[neg] B-ALL cells is a mechanism for CD19[neg] B-ALL relapse after CAR-T cell therapy. A scRNAseq approach can unambiguously detect minor CD19[neg] subclones before CAR-T cell therapy. Thus, analysis of additional B-ALL patients might further demonstrate that such approach represents a useful tool to identify patients at high risk of relapse after CAR-T cell therapy.

## Methods

**Patient's sample.** A child diagnosed with B-ALL was first enrolled in a conventional chemotherapy protocol (CAALL-F01 protocol). During maintenance phase, the patient experienced a frank relapse (relapse 1). Then, anti-CD19 CAR-T therapy was initiated. At day 28 post-CAR-T cells infusion, the patient was in complete remission; however, at day 90, the patient relapsed as noted by an important infiltration of CD10[pos]CD19[neg] leukemic cells in the BM (relapse 2). BM samples were collected before (relapse 1) and after (relapse 2) CAR-T therapy. Informed consent for use of diagnostic and relapse specimens for future research was obtained in accordance with the Declaration of Helsinki. The study was approved by the local ethic committee of "Assistance Publique des Hôpitaux de Marseille".

**Flow cytometry.** BM samples were tested within 24 h of collection. Briefly, following NH4Cl red blood cell lysis, cells were stained with conjugated antibodies (CD45-KrO, CD19-PC7, CD34-ECD, CD10-FITC, CD38-APC, CD58-PC5, CD66c-PE, and CD123-PE) for 30 min at 4 °C and washed twice with FACS buffer (PBS, 2% FCS, and 1 mM EDTA). Data were acquired on Navios cytometer (Beckman Coulter) and analysis was made using Kaluza software (Beckman Coulter). For cell sorting, cells stained with CD19-APC, CD3-PE, and LIVE/DEAD™ Fixable Aqua Dead Cell Stain Kit (Invitrogen) were FACS-sorted using BD-Influx™ cytometer (Becton Dickinson). Antibodies used for flow cytometry are listed in Supplementary Table 2.

**Antibody-HTO conjugation.** Anti-CD45 antibody (clone HI30) provided by BioLegend was irreversibly conjugated to HTO (sequences shown in Supplementary Table 1), using an inverse electron-demand Diels–Alder reaction between methyltetrazine (mTz) and *trans*-cyclooctene (TCO) as previously described[26], and

according to the protocol found at www.cite-seq.com. Briefly, 10 nmol of HTO oligonucleotides were incubated in BBS buffer (50 mM borate, 150 mM NaCl, pH 8.5) containing 10% DMSO and 10 mM TCO-PEG4-NHS (Click Chemistry Tools); while anti-CD45 antibodies were incubated in BBS buffer containing 0.1 mM mTz-PEG4-NHS (Click Chemistry Tools). After 30 min at room temperature, reactions were quenched with 20 mM glycine pH 8.5. TCO-PEG4-HTO and mTz-PEG4-antibodies were then mixed (30 pmol of HTO per 1 μg of antibody) and incubated overnight at 4 °C. At the end of reaction, TCO-PEG4-glycine (1 mM final concentration) was added. TCO-PEG4-glycine was obtained by mixing in water 10 mM TCO-PEG4-NHS with 20 mM glycine pH 8.5. Finally, antibody-HTO samples were concentrated and cleaned-up using an Amicon Ultra-0.5 Centrifugal Filter Unit with 50 kDa cutoff membrane (Merck Millipore). Antibodies-HTO were validated by FACS (Supplementary Fig. 8).

**Single-cell RNA-sequencing experiment.** FACS-sorted cells were labeled with anti-CD45 conjugated to HTO (Supplementary Table 1). Cells were isolated using the Chromium technology from 10× Genomics. Briefly, 5000 cells of each sorted population were pooled, loaded on a Chromium Chip A and droplet-encapsulated with a chromium controller. Single-cell cDNA synthesis and sequencing libraries were prepared with Single-Cell 5′ Library & Gel Bead kit (10× Genomics), according to manufacturer's instructions. Libraries were sequenced using a Nextseq500 and the following parameters, Read1: 26 cycles, i7: 8 cycles, and Read2: 57 cycles.

**ScRNAseq data analysis.** For data preprocessing, mRNA library reads were aligned to the GRCh38 version of the human genome and quantified using Cell-Ranger count (version 3.0.1). Antibody counts for cell hashing were quantified using CITE-seq-count (version 1.4.3, NYGCtech, https://github.com/Hoohm/CITE-seq-Count), using default parameters. The produced mRNA and HTO data matrices were imported into R (v3.5.3), and downstream analysis were performed with the Seurat package (v3.0.1)[22,27].

In order to perform samples demultiplexing, HTOs for each cell were normalized using a centered log ratio transformation across cells. Cells were demultiplexed using the Seurat HTODemux function, and cell doublets and background empty droplets subsequently removed. Before mRNA expression analysis, we filtered out low quality cells from the mRNA matrix (i.e., cells expressing < 200 genes or >10% of mitochondrial-associated genes). Genes expressed in less than three cells were removed. Then, expression raw counts were normalized and scaled using respectively the Seurat NormalizeData and ScaleData functions. The top 2000 variable genes were selected (FindVariableFeatures function; selection method = "vst") to perform a dimensionality reduction using the principal component analysis method (RunPCA function), and a UMAP was computed using the 20 first components (RunUMAP function). We used the Seurat graph-based clustering algorithm to identify clusters of cells (using 20 dimensions and a coarse grained resolution of 0.1). The nonparametric Wilcoxon rank-sum test (FindMarkers seurat fonction) was used for all differential gene expression analysis between two groups of cells.

To investigate large-scale CNV, we used CaSpER tool[23]. Briefly, the scRNAseq experiment bam file was subdivided using 10× Genomics subset-bam, to obtain three bam files, respectively, containing the reads for cells in cluster 0, in cluster 1 and in clusters 2–5. Then BAFExtract was used to calculate BAF for each bam, and finally CaSpER was run using physiological cells (clusters 2–5) as control.

To infer gene regulatory network responsible for clonal transcriptional changes underlying tumoral evolution, we applied SCENIC method[28], using the filtered matrix raw UMI counts.

**Reverse transcription-PCR.** RNA from T1-CD19[pos] and T2-CD19[neg] samples were extracted using RNAeasy mini kit (Qiagen), and bulk cDNA was synthesized using high capacity cDNA reverse transcription kit (Applied Biosystems). Then CD19 transcripts were assessed by PCR using specific primers (Supplementary Table 1) and

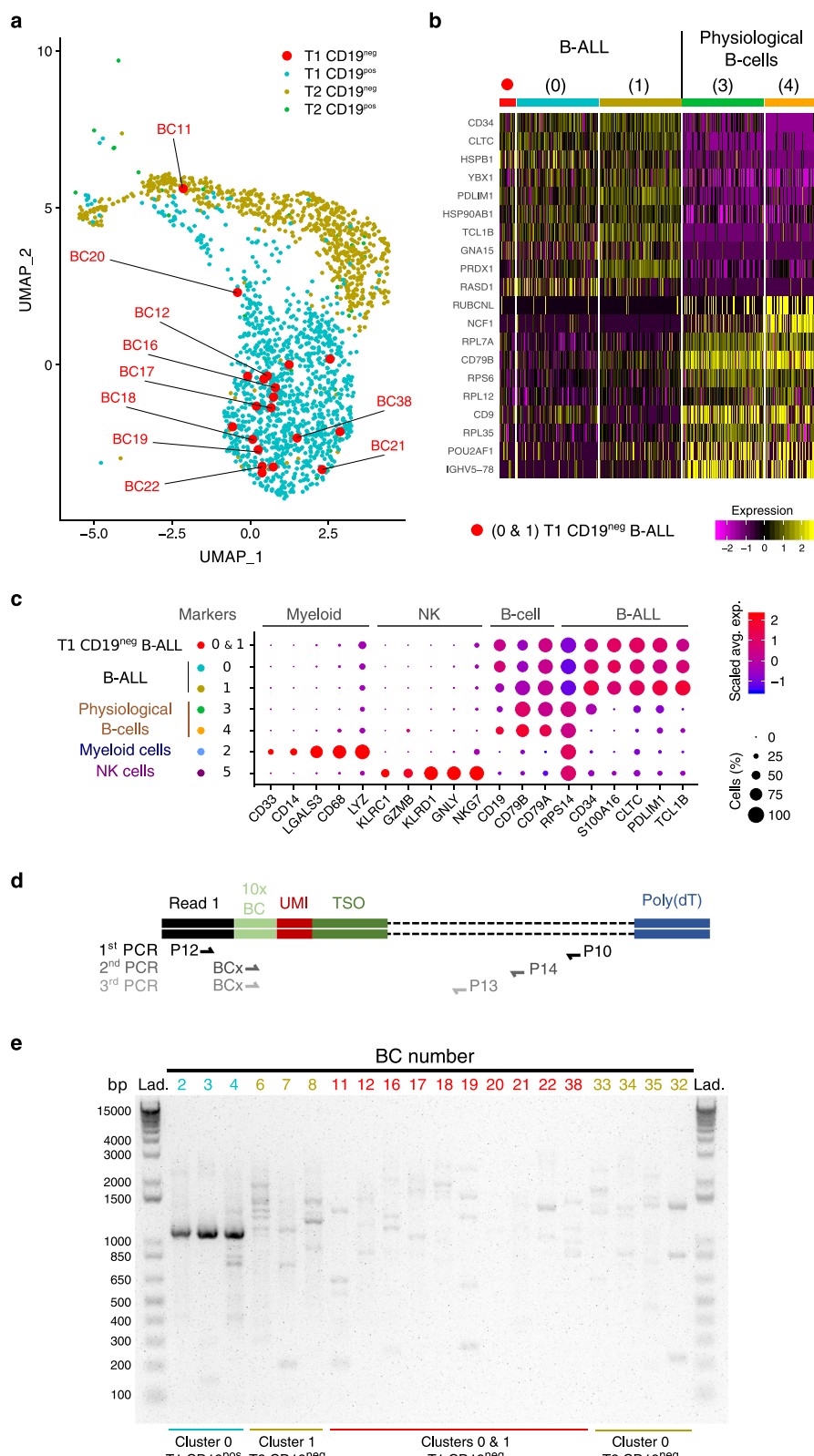

Herculase II enzyme (Agilent technologies). The following PCR program was applied: 95 °C 1 min, 35 cycles (95 °C 20 s, 56 °C 20 s, and 68 °C 90 s), 68 °C 4 min. Sanger sequencing of PCR products was performed by Eurofins Genomics.

**Backtracking of CD19neg clones**. Investigation of CD19 cDNA of individual cells was performed by nested-PCRs using amplified 10× barcoded full-length cDNA. Three successive PCR were carried out with Herculase II enzyme (Agilent technologies) and with the following PCR program: 95 °C 1 min, 20 cycles (95 °C 20 s, 56 °C 20 s, and 68 °C 90 s), 68 °C 4 min (for the third PCR only 15 cycles were performed). In the first PCR, CD19 transcript was amplified with "partial Read1" primer (or P12) and CD19 primer P10. Cell-specific CD19 cDNA amplifications were carried out with primers comprising the 10× cell BC (BC primers) and P14 or P13 for respectively second and third PCR (Supplementary Table 1). For these nested-PCR, 1 µl aliquots of primary or secondary PCR were used. PCR products were analyzed by 1% agarose gel electrophoresis.

**Fig. 2 Detection of CD19[neg] B-ALL clones before CAR-T treatment. a** UMAP plot focused on tumoral cells (clusters 0 and 1) and colored according to sample of origin. The 20 cells from T1-CD19[neg] samples are highlighted by larger red dots. Sequences of 10× cell barcode (BC) of interest are indicated in Supplementary Table 1. **b** Heatmap showing normalized and scaled expression level of 20 genes differentially expressed between tumoral clusters (0 and 1), and either immature B cells (cluster 3) or mature B cells (cluster 4). Gene expression profiles of the 20 T1-CD19[neg] cells are shown independently to the other B-ALL cells. Clusters 0, 1, and 3 were down-sampled to 100 cells for a better readability. Number of expressed genes detected in the various clusters are shown in Supplementary Fig. 10. **c** Dot plot of marker genes expression, same as Fig. 1b except that T1-CD19[neg] cells from clusters 0 and 1 were analyzed independently, and that some differentially expressed genes between tumoral versus physiological cells were added. **d** Nested-PCR strategy to detect CD19 transcript in single cell. **e** Agarose gel of CD19 cDNA amplified by nested-PCR. Lanes are labeled according to the cell BC number for which the sample of origin is indicated at the bottom of the gel. Data are representative of independent backtracking experiments that were performed twice for cells with BC number 32, 33, 34, and 35; and three times for all other cells. Localizations on the UMAP plot of backtracked T2-CD19[neg] cells are shown in Supplementary Fig. 11.

**Reporting summary**. Further information on research design is available in the Nature Research Reporting Summary linked to this article.

## Data availability

Fastq raw data are available in SRA (https://www.ncbi.nlm.nih.gov/sra?term=SRP269742). Processed data can be accessed from the NCBI Gene Expression Omnibus database (accession code GSE153697). Human transcriptome reference used for our analysis is available at 10× Genomics website (https://cf.10xgenomics.com/supp/cell-exp/refdata-cellranger-GRCh38-3.0.0.tar.gz) or in zenodo (https://zenodo.org/record/4114854/files/refdata-cellranger-GRCh38-3.0.0.tar.gz?download=1). The cisTarget database (hg19-tss-centered-10kb-7species.mc9nr.feather) used for SCENIC analysis is available at https://resources.aertslab.org/cistarget/databases/homo_sapiens/hg19/refseq_r45/mc9nr/gene_based/hg19-tss-centered-10kb-7species.mc9nr.feather.

## Code availability

For reproducibility, docker images and detailed scripts used for preprocessing and further analysis are available respectively at zenodo (https://doi.org/10.5281/zenodo.4114854) and github (https://github.com/Delphine-Potier/B-ALL-CAR-T).

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

## Acknowledgements

High-throughput sequencing was performed at the TGML Platform, supported by grants from Inserm, GIS IBiSA, Aix-Marseille Université, and ANR-10-INBS-0009-10. This study was partly supported by research funding from the Canceropôle PACA, Institut National du Cancer and Région Sud. S.P. received funding from the European Union's Horizon 2020 research and innovation program under the Marie Skłodowska-Curie Grant Agreement No. 713750, and from the Regional Council of Provence-Alpes-Côte d'Azur, A*MIDEX (No. ANR-11-IDEX0001-02). D.P.-B. received financial support from ITMO Cancer of AVIESAN (Alliance Nationale pour les Sciences de la Vie et de la Santé, National Alliance for Life Sciences & Health) within the framework of the Cancer Plan (Project No. C19046S) and from CNRS "Osez l'interdisciplinarité!" program-"DMATh" project. The authors acknowledge the "genomic instability and human hemopathies" team and the bioinformatic platform of CIML for technical support.

## Author contributions

T,.R., D.P., M.L., and D.P.-B. performed the experiments, analyzed the data, and wrote the manuscript. D.P., S.P., and M.N. performed bioinformatics analysis. All authors read and approved the final manuscript.

## Competing interests

The authors declare no competing interests.
