## [Peer Review File · Nature Communications]

REVIEWER COMMENTS

Reviewer #1 (Remarks to the Author): with expertise in CD19 CAR-T and B-ALL

Rabilloud et al provide interesting and important data from a single patient relapsing with CD19-negative leukemia after CD19 CAR T cell therapy. This event is relatively frequent and known to be associated with multiple mechanisms including mutations, alternative splicing events as well as post-translational changes. The primary (and still largely unanswered) question is how frequently CD19-negative relapse results from the emergence of pre-existing clones selected by the CD19-targeted immunotherapy or whether CD19 loss is driven by the immunotherapy. Based on analysis of pre- and post- samples isolated based on CD19 expression using RNAseq the authors conclude that this patient's relapsed CD19-negative leukemia emerged from a pre-existing subclone.

Major Comments:

1) A major concern is the extent to which a single patient can be used to generalize about a complex event that is likely to occur via multiple mechanisms. Indeed, leukemic resistance in the context of CD19-targeted immunotherapy can occur in certain subtypes of ALL via conversion to a myeloid phenotype. Thus, the main conclusion of the manuscript would be markedly strengthened by analysis of additional CD19-negative leukemias – which, unfortunately, are relatively common. 2) In the initial publication by Grupp et al in the New England Journal of Medicine reporting on the first three patients receiving CD19 CAR T cells described a CD19 negative relapse emerging from a CD19 negative (by flow cytometry) population present pre-treatment. Admittedly, the analysis performed by Rabilloud et al very elegant and provides additional insights but is not entirely novel.

Reviewer #2 (Remarks to the Author): with expertise in CD19 CAR-T and B-ALL

Synopsis

Using single cell RNA sequencing (scRNASeq) in samples from patients who would relapse with CD19 neg disease the authors identified CD19neg cells prior to anti-CD19 CAR therapy. This was studied in a B-ALL patient who achieved remission post-CAR T cell therapy but relapsed in the marrow as shown by flow cytometry. This also revealed CD19 neg blasts. Single cells CD10+CD19+ and CD19neg cells were purified from samples prior to and following CAR T cell therapy at relapse. scRNAseq identified normal and malignant B cells clustering differently, but also the clustering of pre- and post-CAR T cell treatment leukemic blasts differed in their localization on the UMAP, each with a different transcriptional make-up.

Major comments

This study identifies CD19 negative clones in pre-CAR T cell therapy samples, 20 clones in all, and the gene expression profile of those clones overlaps with those from CD19 neg relapse. Intriguing, but the authors did not dig deeper to convincingly demonstrate that these clones were actually pre-existing. Since the gene expression profiles in T1 and T2 were so different I wonder whether CD19 itself carried any mutations as previously demonstrated by Orlando et al., and whether the mutational spectrum also differed. This study does need a separate validation of the thesis that CD19neg leukemia cells were present prior to therapy

Reviewer #3 (Remarks to the Author): with expertise in scRNA seq and leukemia

This manuscript by Rabilloud et al. reports an interesting case study of an important problem in B-ALL: the emergence of CD19-negative resistance to CD19-directed CAR-T cell therapy. This manuscript is very clear and logically written and deeply investigates a very important question in hematology. Unfortunately, their main assertion, that they can identify pre-existing CD19-negative ALL cells prior to CART therapy, is insufficiently supported by the data.

In this manuscript, the authors compare the single cell (sc) gene expression profiles of CD19-positive and CD19-negative cells collected from a patient with frank ALL prior to receiving CART therapy (T1, first relapse) and after receiving CART therapy (T2, second relapse).

The authors demonstrate that the sc transcriptional profiles of T1 and T2 ALL cells are distinct (populations 0 and 1 in the UMAP plot, Figure 1B) and that the sc transcriptional profiles can be used to assign cells to 4 cellular subsets (B cells, ALL, NK cells, myeloid cells).

Using PCR amplification of bulk samples obtained at T1 and T2, they demonstrate that only an aberrantly spliced transcript of CD19 can be detected at T2 (as previously described)

Finally, by backtracking their barcodes, they demonstrate that 20 (of the 524 sorted CD19-negative cells obtained at T1) map to the T1 CD19-positive, B-ALL region of the UMAP plot (Figure 2A). The basis for their claim, that they can detect pre-existing, CD19-negative B-ALL cells lies in the detection of these 20 T1 CD19-negative cells in their scRNA seq data. I have two major concerns with this conclusion:

1. It is unclear what the specificity of the UMAP method is. These methods are notoriously error prone. That is to say, do we have any reason to believe that these 20 cells are truly ALL cells and not mis-assigned? For example, in Figure 1D, I see some T1 CD19+ cells (presumably these would be ALL cells) assigned to the myeloid and NK cell clusters.
2. The 20 T1 CD19- cells represent about 4% of the total number of T1 cells captured for RNA seq (according to the barcode deconvolution: there were 524 cells sequenced from this population, 20 of them were assigned to the ALL cluster). However, the flow plots identified only 0.03% of possible cells that could qualify as CD19-negative ALL at T1. These data suggest that the 4% of T1 CD19negative cells identified as ALL are more likely mis-assigned than true ALL cells.

This manuscript might be more impactful if:

1. The gene expression profiles of the 20 T1 CD19 cells that are claimed to be ALL should be compared more specifically to ALL AND non ALL cells to understand their basis of their computational similarity to ALL: is there anything in the gene expression that shows them to be more likely ALL and NOT likely something else? Do these cells express sufficient number of genes to convince us that their gene expression profiles are reliable indicators of their identity?
2. If the T1 CD19negative cells could be validated as truly ALL, rather than mis-assigned non-ALL cells, it would be very interesting to compare the gene expression profiles of these 20 cells to the remaining ALL cells at T1. Are there genes, beyond CD19, that might hint at a mechanism of resistance? Why is CD19 mis-spliced in these cells? etc.

REVIEWER COMMENTS

Reviewer #1 (Remarks to the Author): with expertise in CD19 CAR-T and B-ALL

Rabilloud et al provide interesting and important data from a single patient relapsing with CD19-negative leukemia after CD19 CAR T cell therapy. This event is relatively frequent and known to be associated with multiple mechanisms including mutations, alternative splicing events as well as post-translational changes. The primary (and still largely unanswered) question is how frequently CD19-negative relapse results from the emergence of pre-existing clones selected by the CD19-targeted immunotherapy or whether CD19 loss is driven by the immunotherapy. Based on analysis of pre- and post- samples isolated based on CD19 expression using RNAseq the authors conclude that this patient's relapsed CD19-negative leukemia emerged from a pre-existing subclone.

Major Comments:

1) A major concern is the extent to which a single patient can be used to generalize about a complex event that is likely to occur via multiple mechanisms. Indeed, leukemic resistance in the context of CD19-targeted immunotherapy can occur in certain subtypes of ALL via conversion to a myeloid phenotype. Thus, the main conclusion of the manuscript would be markedly strengthened by analysis of additional CD19-negative leukemias – which, unfortunately, are relatively common.

Currently we do not have any supplementary samples from CAR-T treated patients who relapsed as CD19^{neg} B-ALL. Nevertheless, we fully agree with the reviewer that additional samples would have been a plus for our study; we stressed this point by modifying the last sentence of the discussion: 'scRNAseq approach can unambiguously detect minor CD19^{neg} subclones before CAR-T cell therapy. Thus, analysis of additional B-ALL patients might further demonstrate that such approach represents a useful tool to identify patients at high-risk of relapse after CAR-T cell therapy'.

2) In the initial publication by Grupp et al in the New England Journal of Medicine reporting on the first three patients receiving CD19 CAR T cells described a CD19 negative relapse emerging from a CD19 negative (by flow cytometry) population present pre-treatment. Admittedly, the analysis performed by Rabilloud et al very elegant and provides additional insights but is not entirely novel.

The initial paper from Grupp et al. (ref#14) describes the first B-ALL patients treated with anti-CD19 CAR-T cells and shows the emergence of CD19^{neg} blasts in one patient. The authors did not clearly state nor demonstrate that CD19^{neg} cells detected by flow cytometry, before CAR-T treatment, were leukemic. Indeed, the gating strategy used (CD19^{neg}, CD45^{pos}, SSC low, and CD34^{pos}) is insufficient to prove that this population is B-ALL. In fact, Grupp et al. prove the existence of CD19^{neg} B-ALL clones at day 23 post-CAR-T infusion and not before CAR-T treatment.

Reviewer #2 (Remarks to the Author): with expertise in CD19 CAR-T and B-ALL

Synopsis

Using single cell RNA sequencing (scRNASeq) in samples from patients who would relapse with CD19^{neg} disease the authors identified CD19^{ng} cells prior to anti-CD19 CAR therapy. This was studied in a B-ALL patient who achieved remission post-CAR T cell therapy but

relapsed in the marrow as shown by flow cytometry. This also revealed CD19 neg blasts. Single cells CD10+CD19+ and CD19neg cells were purified from samples prior to and following CAR T cell therapy at relapse. scRNAseq identified normal and malignant B cells clustering differently, but also the clustering of pre- and post-CAR T cell treatment leukemic blasts differed in their localization on the UMAP, each with a different transcriptional make-up.

Major comments

This study identifies CD19 negative clones in pre-CAR T cell therapy samples, 20 clones in all, and the gene expression profile of those clones overlaps with those from CD19 neg relapse. Intriguing, but the authors did not dig deeper to convincingly demonstrate that these clones were actually pre-existing. Since the gene expression profiles in T1 and T2 were so different I wonder whether CD19 itself carried any mutations as previously demonstrated by Orlando et al., and whether the mutational spectrum also differed. This study does need a separate validation of the thesis that CD19neg leukemia cells were present prior to therapy

CD19 mutation. Indeed, Orlando et al (ref#4) identified in 12 CD19-negative relapsed patients, mutations throughout exons 2-5 of CD19. For both samples harvested at T1 and T2, our analysis of CD19 sequencing reads did not reveal any somatic mutation within exons 2 to 5. Yet, we showed that loss of CD19 is due to the expression of a non-functional CD19 isoform retaining intron 2. Unfortunately, we could not find an obvious explanation to this altered splicing of CD19. Of note, the study from Asnani et al (ref#6) was focused on this non-functional CD19 isoform and they notably showed that this mis-splicing was recurrent in B-ALL samples. However, the mechanism of this alternative splicing was not identified, suggesting that this mechanism might be complex and probably requires to be investigated in a dedicated study. We stressed this point in the discussion of the revised version.

Relation between T1 and T2 ALL. CD19-negative B-ALL cells are clearly related to CD19-positive B-ALL clones as shown by CNV events analysis using CaSpER and B-allele frequency (BAF) approaches (new supplementary fig. 2).

At the relapse (T2), the transcriptional profile slightly evolves, thus most of T2-CD19^{neg} tumoral cells were found in cluster 1, yet some of them were detected within cluster 0 (new supplementary fig. 8) and backtracking experiments further confirmed that those cells were CD19 negative (now shown in the extended gel of fig. 2E). Among the most differentially expressed genes between T1 and T2, we pointed out KLF6. Of note, BAF analysis did not reveal any loss of KLF6 allele. Although KLF6 expression can be regulated by external stimuli [Slavin et al. Oncogene, **23** (2004)], we feel that suggesting a role of the microenvironment on KLF6 expression is too speculative. Moreover, the identification of the mechanism leading to KLF6 down-regulation is out of scope of our manuscript.

Validation of pre-existing CD19 negative B-ALL

We agree with both reviewers 2 & 3 who recommended to better prove the leukemic origin of T1 CD19-negative cells. The comparisons of gene expression profiles (Fig. 2: new panel B; and new supplementary Fig. 5) demonstrate that the 20 CD19-negative cells detected before CAR-T therapy (T1) are true B-ALL. Moreover, CaSpER and SNP analyses of our scRNAseq data indicate that like the other T1/T2 B-ALL cells (and in contrast to non-B-ALL cells), those T1-CD19^{neg} cells, harbor genomic deletions in chromosomes 9 and 5 (new supplementary fig. 2).

Reviewer #3 (Remarks to the Author): with expertise in scRNA seq and leukemia

This manuscript by Rabilloud et al. reports an interesting case study of an important problem in B-ALL: the emergence of CD19-negative resistance to CD19-directed CAR-T cell therapy. This manuscript is very clear and logically written and deeply investigates a very important question in hematology. Unfortunately, their main assertion, that they can identify pre-existing CD19-negative ALL cells prior to CART therapy, is insufficiently supported by the data.

In this manuscript, the authors compare the single cell (sc) gene expression profiles of CD19-positive and CD19-negative cells collected from a patient with frank ALL prior to receiving CART therapy (T1, first relapse) and after receiving CART therapy (T2, second relapse).

The authors demonstrate that the sc transcriptional profiles of T1 and T2 ALL cells are distinct (populations 0 and 1 in the UMAP plot, Figure 1B) and that the sc transcriptional profiles can be used to assign cells to 4 cellular subsets (B cells, ALL, NK cells, myeloid cells).

Using PCR amplification of bulk samples obtained at T1 and T2, they demonstrate that only an aberrantly spliced transcript of CD19 can be detected at T2 (as previously described)

Finally, by backtracking their barcodes, they demonstrate that 20 (of the 524 sorted CD19-negative cells obtained at T1) map to the T1 CD19-positive, B-ALL region of the UMAP plot (Figure 2A). The basis for their claim, that they can detect pre-existing, CD19-negative B-ALL cells lies in the detection of these 20 T1 CD19-negative cells in their scRNA seq data. I have two major concerns with this conclusion:

1. It is unclear what the specificity of the UMAP method is. These methods are notoriously error prone. That is to say, do we have any reason to believe that these 20 cells are truly ALL cells and not mis-assigned? For example, in Figure 1D, I see some T1 CD19+ cells (presumably these would be ALL cells) assigned to the myeloid and NK cell clusters.

We fully agree with the reviewer that we should better prove that the 20 T1-CD19^{neg} cells detected before CAR-T treatment were not mis-assigned and that they are true B-ALL.

Thus, as suggested in the reviewer's first recommendation below, we performed the gene expression comparison of B-ALL cells (excluding the 20 T1 CD19^{neg} cells to avoid any bias) versus physiological B-cells, myeloid or NK cells. The resulting heatmaps displayed in new fig. 2B and new supplementary fig. 5, show that gene expression profiles of these 20 T1-CD19^{neg} cells and the other B-ALL cells are similar.

Moreover, CNV events prediction using the recently developed CaSpER tool (ref#11) and B-allele frequency (BAF) approaches indicate that like the other T1/T2 B-ALL cells (and in contrast to non-B-ALL cells), T1-CD19^{neg} population harbor genomic deletions in chromosomes 9 and 5 (new supplementary fig. 2).

2. The 20 T1 CD19- cells represent about 4% of the total number of T1 cells captured for RNA seq (according to the barcode deconvolution: there were 524 cells sequenced from this population, 20 of them were assigned to the ALL cluster). However, the flow plots identified only 0.03% of possible cells that could qualify as CD19-negative ALL at T1. These data suggest that the 4% of T1 CD19negative cells identified as ALL are more likely mis-assigned than true ALL cells.

Indeed, using FACS we estimated that around **0,03% of live cells** could qualify as CD19^{neg} B-ALL at T1. ScRNAseq data indicate that within the 524 T1-CD19^{neg} cells, 20 cells correspond to B-ALL cells, thus these cells represent 3,8% of T1-CD19^{neg} cells. However, T1-CD19^{neg} cells represent only 3% of live cells (Fig. 1A) therefore the 20 T1-CD19^{neg} B-ALL cells represent around **0,1% of live cells** (which is in the range of our FACS estimation).

This manuscript might be more impactful if:

1. The gene expression profiles of the 20 T1 CD19 cells that are claimed to be ALL should be compared more specifically to ALL AND non ALL cells to understand their basis of their computational similarity to ALL: is there anything in the gene expression that shows them to be more likely ALL and NOT likely something else? Do these cells express sufficient number of genes to convince us that their gene expression profiles are reliable indicators of their identity?

As replied for the main point 1 above, we performed differential gene expression analysis of B-ALL versus non-B-ALL cells, and this showed that gene expression profile of the 20 T1-CD19^{neg} is similar to leukemic cells (new fig. 2B and supplementary fig. 5). Also, in fig. 2C, the new dot plot reports expression of marker genes in T1-CD19^{neg} cells.

A violin plot showing the number of expressed genes detected in the various cell types, is now displayed in new supplementary fig. 7. T1-CD19^{neg} cells express around 2090 genes/cell, which is sufficient to reliably assess their identity.

2. If the T1 CD19negative cells could be validated as truly ALL, rather than mis-assigned non-ALL cells, it would be very interesting to compare the gene expression profiles of these 20 cells to the remaining ALL cells at T1. Are there genes, beyond CD19, that might hint at a mechanism of resistance? Why is CD19 mis-spliced in these cells? etc.

We attempted to compare the 20 T1 CD19-negative cells to the remaining T1 ALL cells. Yet we did not find any obvious differences (see below the table of the top “differentially expressed” genes). None of those genes is involved in mRNA splicing. Unfortunately, we could not find an obvious explanation to CD19 mis-splicing. Of note, the study from Asnani et al (ref#6) was focused on this non-functional CD19 isoform and they notably showed that this mis-splicing was recurrent in B-ALL samples. However, no clear mechanism for intron 2 retention was identified, suggesting that the mechanism might be complex and probably requires to be investigated in a dedicated study. We stressed this point in the discussion of the revised version.

Gene	P_val	Avg_logFC	T1 CD9neg	T1 CD19pos	P_val_adj
SPAST	0.0000013	0.29	0.30	0.05	0.0025
SMYD2	0.000017	0.31	0.30	0.06	0.33
HOOK2	0.00020	0.40	0.50	0.20	1.0
RBM42	0.00057	-0.50	0.00	0.39	1.0
RNF122	0.00064	0.30	0.40	0.14	1.0

REVIEWER COMMENTS

Reviewer #2 (Remarks to the Author):

The thesis that CD19neg cells are pre-existent in patients who ultimately relapse with CD19 neg dz is an interesting one and along similar lines in other drug resistance observations. This study is still not convincing enough and should have performed mechanistic follow-up experiments to, for instance, conclusively demonstrate that the retained intron 2 was indeed responsible for a lack of CD19 expression. The reference errors, mixing the Grupp 2013 NEJM paper up with Kalos' 2011 CLL paper, is odd as well. That aside, I am not convinced that they have proven their point.

Reviewer #3 (Remarks to the Author):

In this resubmission, Rabilloud demonstrate that the T1 CD19negative cells are indeed ALL, primarily as evidenced by the gene expression profiles of these cells (Figure 2B) and confirmation that these cells express adequate transcripts to be accurately categorized (Supplemental Figure 7). This data is highly convincing.

While this is only a single patient case study, it is highly informative and addresses an unanswered question in the field.

I find the data as presented in this revision convincing and the conclusions of the authors supported by the data. I only have minor comments at this time in regards to the data in Supplemental Figure 2 and in regards to the clarity of some of the figures. Overall, I find the additional gene expression analysis sufficient to address my original concerns about the strength of their conclusions.

Minor Comments

- Supplemental figure 2A: chr 9 loss in T1 and T2 ALL demonstrates that both the T1 and T2 cells arise from the same original leukemia. This is not the question and also this can be easily shown by traditional karyotype analysis. The question is whether the CD19neg T1 cells are preexisting. It would be useful to define whether the 20 CD19neg T1 cells that cluster with the T1 ALL cells in the Figure 1D UMAP plot have the same chr9 deletion seen in the ALL; to see the chr9 map of the 20 T1-CD19- cells that cluster with the T1 ALL cells, even if it is only found in 50% of the cells (presumably, this is due to technical issues?). Why is that data not shown?
- Supplemental Figure 2B: In this plot we see 1 chr5 SNP that is informative in the T1 CD19neg cells and this shows a pattern consistent with ALL (loss of 1 copy of that SNP). Are these 2 SNPs the only SNPs that are covered on Chr5?

Suggestions to improve clarity/readability of the manuscript:

- Figure 1B: requires better explanation: what does the size of the circles represent, % of what? What does "expression value" represent? Given that there are negative numbers on the legend, I am assuming this is some sort of Z-score? That needs to be clarified. Are the NK and myeloid cells mislabeled on this plot?
- Supple Figure 2B: The top row is annotated as "(0) B-ALL (Others samples)." I don't know what the "Others Samples" means. Presumably the cluster annotation (0-5) corresponds to the clusters in figure 1.
- Suppl Figure 2B: It would be helpful to explain the color coding (green/brown, red/blue) in a legend

REVIEWER COMMENTS

Reviewer #1: (not contacted)

Reviewer #2 (Remarks to the Author):

The thesis that CD19neg cells are pre-existent in patients who ultimately relapse with CD19 neg dz is an interesting one and along similar lines in other drug resistance observations. This study is still not convincing enough and should have performed mechanistic follow-up experiments to, for instance, conclusively demonstrate that the retained intron 2 was indeed responsible for a lack of CD19 expression. The reference errors, mixing the Grupp 2013 NEJM paper up with Kalos' 2011 CLL paper, is odd as well. That aside, I am not convinced that they have proven their point.

The aim of our study is to determine whether a CAR-T cell-treated B-ALL patient who relapsed as a CD19-negative B-ALL, carried CD19-negative leukemic clones before CAR-T therapy. Using state-of-the-art approaches, we manage to detect and thus prove the existence of such clones. We do not clearly understand why the reviewer is not convinced by our data.

In our study, we showed that at the relapse only the transcript isoform retaining CD19 intron 2 is detected; *de facto* this will introduce a stop codon impeding the expression of CD19 protein. Yet, as mentioned in our first response to reviewers' comments (and as indicated in the discussion of our manuscript) characterization of the mechanism leading to inhibition of CD19 expression is out of scope of this study.

We cited Grupp et al. (ref #14) in the discussion. To be more precise, we discussed the results shown in the figure 3 of Grupp et al. paper. Thus, we definitely did not mix up Grupp et al. NEJM paper with any other paper.

Reviewer #3 (Remarks to the Author):

In this resubmission, Rabilloud demonstrate that the T1 CD19negative cells are indeed ALL, primarily as evidenced by the gene expression profiles of these cells (Figure 2B) and confirmation that these cells express adequate transcripts to be accurately categorized (Supplemental Figure 7). This data is highly convincing.

While this is only a single patient case study, it is highly informative and addresses an unanswered question in the field.

I find the data as presented in this revision convincing and the conclusions of the authors supported by the data. I only have minor comments at this time in regards to the data in Supplemental Figure 2 and in regards to the clarity of some of the figures. Overall, I find the additional gene expression analysis sufficient to address my original concerns about the strength of their conclusions.

We deeply thank the referee for his constructive and benevolent comments all along this reviewing process. His contribution helped us to improve significantly our manuscript.

Minor Comments

- Supplemental figure 2A: chr 9 loss in T1 and T2 ALL demonstrates that both the T1 and T2 cells arise from the same original leukemia. This is not the question and also this can be easily shown by traditional karyotype analysis. The question is whether the CD19neg T1 cells are preexisting. It would be useful to define whether the 20 CD19neg T1 cells that cluster with the T1 ALL cells in the Figure 1D UMAP plot have the same chr9 deletion seen in the ALL; to see the chr9 map of the 20 T1-CD19- cells that cluster with the T1 ALL cells, even if it is only found in 50% of the cells (presumably, this is due to technical issues?). Why is that data not shown?

The 20 T1 CD19neg cells are now clearly shown in Supplementary Fig 2A.

Designers of CaSpER algorithm (Serin Harmanci et al. ref#11) evaluated the performance of their tool to detect correctly CNV by calculating true positive rate (TPR) and false positive rate (FPR). In their analysis of single-cell RNA-seq data, Serin Harmanci et al. indicate that CaSpER achieves 45% TPR and 3% FPR in deletion events. Thus, we did not expect that our CaSpER analysis successfully detects 9q deletion in all the B-ALL cells. As described in Serin Harmanci et al., CaSpER's user can set up the γ parameter threshold (between 1 to 9) to tune the tradeoff between FPR and TPR rates. Thus, the most stringent condition, $\gamma = 9$, is more specific (lower FPR but also lower FDR); while the most relax condition, $\gamma = 1$, is more sensitive (higher FDR) but it is also expected to be less specific (higher FPR). In the analysis shown in supplementary Fig. 2A we were very stringent (CaSpER was run with $\gamma = 9$); we favored specificity over sensitivity. However, if we use a slightly more relax parameter $\gamma = 5$, then 9q deletion is found in 16 out of the 20 T1 CD19neg (as shown in the figure on the right, the 20 T1 CD19neg are represented by larger dots circled in red).

- Supplemental Figure 2B: In this plot we see 1 chr5 SNP that is informative in the T1 CD19neg cells and this shows a pattern consistent with ALL (loss of 1 copy of that SNP). Are these 2 SNPs the only SNPs that are covered on Chr5?

Indeed, within the altered 5q31.2-5q33.2 region the 4 SNP shown in Supplementary Fig. 2B are the only ones which were informative : heterozygous and with enough coverage (>100 reads) in the 3 groups (cluster 0, cluster 1 and physiologic clusters 2, 3, 4 & 5).

Suggestions to improve clarity/readability of the manuscript:

- Figure 1B: requires better explanation: what does the size of the circles represent, % of what? What does “expression value” represent? Given that there are negative numbers on the legend, I am assuming this is some sort of Z-score? That needs to be clarified. Are the NK and myeloid cells mislabeled on this plot?

We agree with the reviewer and improved the labelling of figure 1B as well as its legend.

‘(B) Seurat Dotplot showing the expression level of marker genes in each cluster. Dot size represents the percentage of cell expressing the gene of interest, while dot color represents the scaled average expression (Scaled Avg. Exp.) of the gene of interest across the various clusters (a negative value corresponds to an expression below the mean expression).’

We deeply thank the reviewer who noticed that, by mistake, we inverted NK and myeloid labels in plots Fig 1B (and Fig 2C).

- Supple Figure 2B: The top row is annotated as “(0) B-ALL (Others samples).” I don’t know what the “Others Samples” means. Presumably the cluster annotation (0-5) corresponds to the clusters in figure 1.

We agree, our annotation of Supplementary Fig 2B was not quite clear. The annotation is now more precise; also, the legend is more explicit.

- Suppl Figure 2B: It would be helpful to explain the color coding (green/brown, red/blue) in a legend

Color codes are now indicated in the Supplementary Fig 2 legend.

REVIEWERS' COMMENTS

Reviewer #3 (Remarks to the Author):

The authors have very thoroughly addressed all of my concerns and I congratulate them on this interesting manuscript.